# Flexural Behavior of One-Way Slab Reinforced with Grid-Type Carbon Fiber Reinforced Plastics of Various Geometric and Physical Properties

**Kyung-Min Kim * and Ju-Hyun Cheon**

Construction Technology Research Center, Korea Conformity Laboratories (KCL), 199, Gasan Digital 1-ro, Geumcheon-gu, Seoul 08503, Republic of Korea
* Correspondence: kymkim@kcl.re.kr; Tel.: +82-2-2102-2755

**Abstract:** Textile-reinforced concrete (TRC) has many advantages, including corrosion resistance, but TRC is a novel composite material and there is limited experimental research on the flexural behavior of TRC members. This paper aims to experimentally evaluate the flexural behavior of TRC slabs reinforced with nine types of grid-type carbon fiber-reinforced plastic (CFRP) (hereafter referred to as carbon grid) with varying cross-sectional areas, spacings, tensile strengths, and elastic moduli of longitudinal strands. The experimental results show that the maximum load tends to be higher in specimens reinforced with carbon grids with small cross-sectional areas and spacings of strands but high tensile strength. Cross-sectional area and spacing were also revealed to influence the crack-formation stage behavior. On the other hand, stiffness decreased to approximately 8% or lower than the initial stiffness, with cracking in all carbon grid-reinforced specimens; post-peak behavior also exhibited dependency on tensile stress acting on the carbon grids under the maximum load, based on 80% of the tensile strength.

**Keywords:** textile-reinforced concrete; carbon grid; flexural behavior; CFRP reinforcement





## 1. Introduction

Reinforced concrete (RC) is the most popular composite material in structures since the 20th century owing to its economic efficiency, availability, and mechanical properties. However, exposure to a marine environment may rapidly degrade the durability of RC structures as the reinforcing bars are constantly corroded by chloride attacks [1,2]. To prevent such chloride attacks, certain design regulations are followed to maintain the minimum cover depth for RC members, and the void between concrete and reinforcing bars is minimized via effective construction.

Furthermore, to coat the surface of reinforcing bars with epoxy, a technology was developed [3,4] to prevent the chloride ions from corroding the bars; however, this technology was applied only to a limited number of bridges owing to the decreased coating effect and the occurrence of local corrosion when the coating layer is peeled off or torn during concrete pouring [5]. Moreover, the surface epoxy coating may lead to bond problems between the bars and concrete [6].

Since the 1970s, several studies considered the application of fiber-reinforced plastics (FRPs) with a strong resistance to corrosion when used as the reinforcement for concrete structures, rather than using reinforcing bars, to prevent chloride attacks on RC structures [7]. FRP reinforcement was primarily applied to bridge slabs in the form of bars, using glass, carbon, and aramid fibers as reinforcing fibers [7,8]. Recently, the development of grid-type FRP reinforcement [9] and FRP tendons [10] led to the expansion of applying FRP reinforcement to various structures, including building walls [11], external strengthening of structures [12,13], and bridge foundations.

Among them, the grid-type FRP reinforcement is a type of textile reinforcement for textile-reinforced concrete (TRC) structures [14] that uses fibers woven in grid form as the reinforcement for concrete. Typical textile reinforcement is produced by only weaving fibers, such as glass, aramid, carbon, and basalt fibers, in grid form without impregnating them with resin. Although the grid-type FRP reinforcement exhibits higher tensile strength than reinforcing bars, the absence of plastic deformation capacity results in rapid brittle fracture after reaching the tensile strength [7]. Several researchers [15,16] reported that when fibers are impregnated or coated with resin such as FRP reinforcement, the tensile behavior of TRC members is improved in terms of smaller crack spaces, enhanced stable crack geometry, and increased maximum load.

The flexural behavior of TRC members differs from that of RC members [17] because textile reinforcement with physical properties different from reinforcing bars are used as the reinforcement for concrete (Figure 1). In other words, a linear elastic behavior is observed, similar to that in RC members, before the occurrence of cracks. However, the continuous occurrence of multiple cracks results in the transformation of tensile stress to textile reinforcement at the crack cross-section; no new cracks occur after the crack formation stage, where the load hardly increases. As cracking is stabilized and textile reinforcement resists the load, the load gradually increases owing to the increase in deformation, leading to failure because of the non-homogeneous pull-out between the fibers that constitute the textile reinforcement or the fracture of the textile reinforcement.

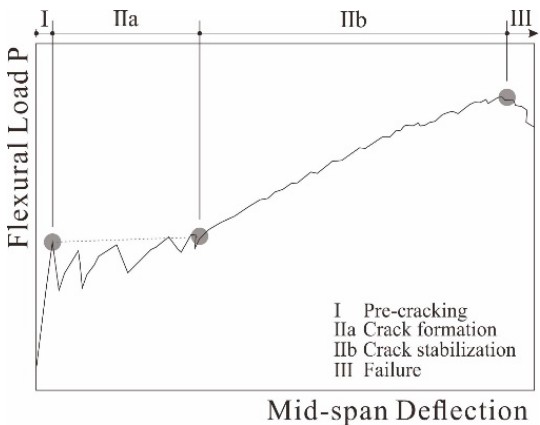

**Figure 1.** Load–deflection of mid-span relationship for a textile-reinforced concrete (TRC) one-way slab.

The flexural behavior of TRC members was analyzed considering the textile reinforcement that uses fibers not impregnated with resin. However, only a few studies evaluated the flexural behavior of TRC members reinforced with grid-type FRP reinforcement that impregnates fibers with resin [18–20]. These studies also used grid-type FRP reinforcement with a small strand spacing of less than 50 mm in the weft and warp directions and a small strand cross-sectional area because of the high tensile strength of FRPs.

In this study, the flexural behavior of TRC one-way slabs was experimentally evaluated by applying grid-type carbon fiber-reinforced plastic (CFRP) as a reinforcement for concrete, hereafter referred to as carbon grid, manufactured by impregnating carbon fibers with resin. Nine types of carbon grids, manufactured by two different methods and with different geometric and physical properties, were used. The flexural behavior of TRC slabs was also directly compared to that of RC slabs. To this end, one-way slab specimens were prepared by applying carbon grids using two different manufacturing methods with varying cross-sectional areas, spacings, tensile strengths, and elastic moduli of longitudinal strands and reinforcing bars. Their flexural behaviors were evaluated using a three-point bending test.

## 2. Experimental Design

### 2.1. Specimen Overview

Nine one-way slab specimens reinforced with carbon grids were prepared to evaluate their flexural behavior; Table 1 summarizes the specifications of different parameters of the specimens. The primary experimental variables were the cross-sectional area (1.7, 1.8, 3.5, 17.5, 26.4, 65, and 100 mm²), spacing (21, 38, 50, and 100 mm), tensile strength (1200, 1400, 3600, and 4000 MPa), and elastic modulus (100, 165, and 230 GPa) of strands according to the carbon grid-manufacturing method (biaxial warp-knitted and cross-laminate structures). For comparison, a one-way slab specimen reinforced with steel was also prepared. Typically, carbon grids are chemically stable [7] and exhibit high resistance to corrosion caused by chloride attacks; consequently, a minimum cover depth is not necessary to prevent chloride attacks [21]. Therefore, a cover depth of less than 10 mm was set for the specimens reinforced with carbon grids in this study. The size of the carbon grid-reinforced specimens (Figure 2a–c) was determined so that the failure mode was governed by mortar crushing considering the universal testing machine (UTM) condition. The size of the steel-reinforced specimen (Figure 2d) was also determined so that the tensile capacity by reinforcement was similar to the specimens reinforced with both the biaxial warp knitted and cross-laminate structured carbon grids.

**Table 1.** Specifications of the one-way slab specimens.

| Specimen | Width (mm) | Depth (mm) | Length (mm) | Carbon Grid | | |
|---|---|---|---|---|---|---|
| | | | | Name | Number of Longitudinal Direction Grids | Maximum Tensile Load by Grids or Steel (kN) |
| G1 | | | | Q95 | 8 | 101.3 |
| G2 | | | | Q85 | 15 | 106.3 |
| G3 | | | | Q47 | 8 | 50.1 |
| J1 | | | | FTG-CR6 | 3 | 73.5 |
| J2 | 300 | 50 | 1000 | FTG-CR8_50 | 5 | 184.8 |
| J3 | | | | FTG-CR8 | 3 | 110.9 |
| J4 | | | | FTG-CR13 | 3 | 273.0 |
| J5 | | | | FTG-CR16 | 3 | 420.0 |
| J6 | | | | FTG-CMR16 | 3 | 360.0 |
| RC | | 65 | | (Steel, D10) | 3 | 85.6 |

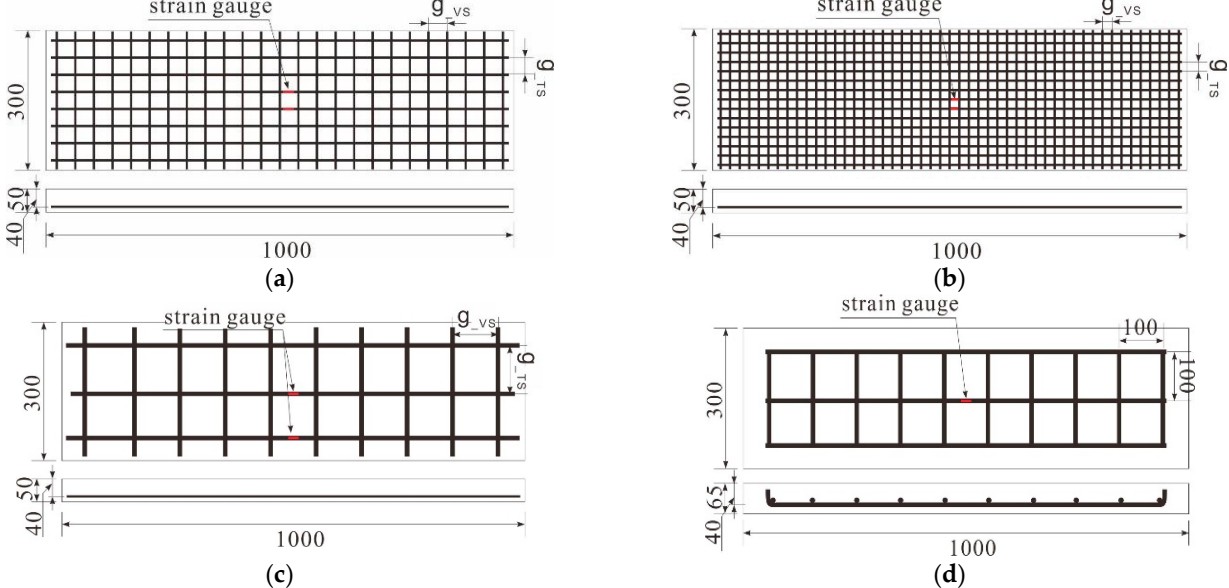

**Figure 2.** Specimens reinforced by: (**a**) Q95 and Q47; (**b**) Q85; (**c**) FTG-CR6, FTG-CR8_50, FTG-CR8, FTG-CR13, FTG-CR16, and FTG-CMR, and (**d**) steel.

Two types of structures, produced by different manufacturing methods, were observed in the carbon grids. The first was a biaxial warp-knitted structure, hardened by epoxy resin impregnation. Herein, the carbon fiber strands were pulled in the weft and warp directions, and separate fibers were woven in the warp direction (Figure 3a) [16]. Conversely, a cross-laminate structured carbon grid was manufactured by laminating thin strips of carbon fibers in the weft and warp directions and hardening them by vinyl ester resin impregnation (Figure 3b).

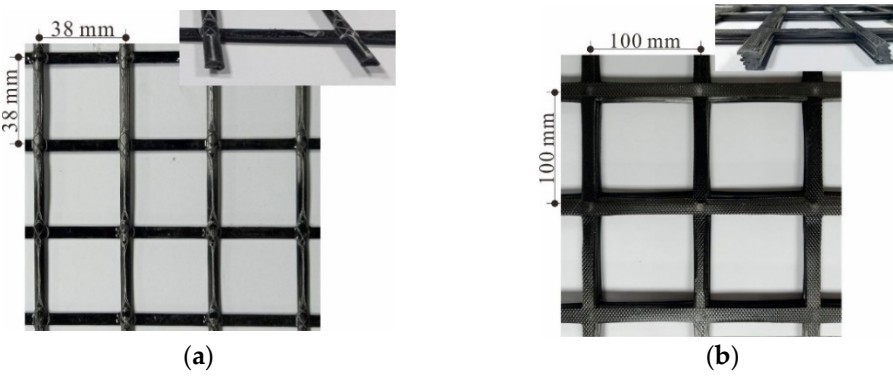

(a)　　　　　　　　　　　　　　　　　　　　　(b)

**Figure 3.** Carbon grids with: (**a**) biaxial warp-knitted; and (**b**) cross-laminate structures.

Table 2 lists the primary characteristics of these carbon grids. The biaxial warp-knitted structured carbon grid exhibited a higher tensile strength and elastic modulus than the cross-laminate structured carbon grid; however, the spacing and cross-sectional area of strands was smaller in the former. On the other hand, SD400-D10 was used in the steel-reinforced specimen that was compared to the carbon grid-reinforced specimens, and its tensile properties are presented in Table 2.

**Table 2.** Characteristics of the carbon grids and tensile properties.

| Name | Structure | Resin | Area (mm²/EA) | Grid Geometry (Longitudinal ($G_{VS}$) × Transverse ($G_{TS}$) Spacing, mm) | Tensile Strength (MPa) | Tensile Modulus of Elasticity (GPa) |
|---|---|---|---|---|---|---|
| Q47 | Biaxial warp-knitted structure | Epoxy | 1.7 | 38 × 38 | 3600 | 230 |
| Q85 | | | 1.8 | 21 × 21 | 4000 | |
| Q95 | | | 3.5 | 38 × 38 | 3600 | |
| FTG-CR8_50 | Cross-laminate structure | Vinyl Ester | 26.4 | 50 × 50 | 1400 | 100 |
| FTG-CR6 | | | 17.5 | | | |
| FTG-CR8 | | | 26.4 | | | |
| FTG-CR13 | | | 65.0 | 100 × 100 | | |
| FTG-CR16 | | | 100.0 | | | |
| FTG-CMR16 | | | | | 1200 | 165 |
| D10 | - | - | - | - | 553 (452 *) | 177 |

* Yield strength of steel, D10.

The strand spacings of the carbon grids were 21, 38, 50, and 100 mm; except for the 100-mm-spacing, concrete with coarse aggregate could not be appropriately poured between strands. Therefore, the specimens were prepared using mortar, which did not use coarse aggregate [20,22]. Table 3 presents the mortar mix design and the compressive strength of mortar at 28 days. In the case of specimens reinforced with the carbon grids, the mortar was initially poured to the height corresponding to the cover thickness. Subsequently, the carbon grids were placed, and the mortar was again poured up to the height of the specimens (Figure 4). Vibration compaction was also conducted during mortar pouring.

The mold was removed after one day, and the specimens were cured in a laboratory at a temperature of 20 °C and 60% humidity.

**Table 3.** Mix design and compressive strength of mortar.

| Cement (kg/m³) | Water (kg/m³) | W/C (%) | Fine Aggregate (kg/m³) | Compressive Strength (MPa) |
|---|---|---|---|---|
| 450 | 171 | 38 | 1485 | 30.4 |

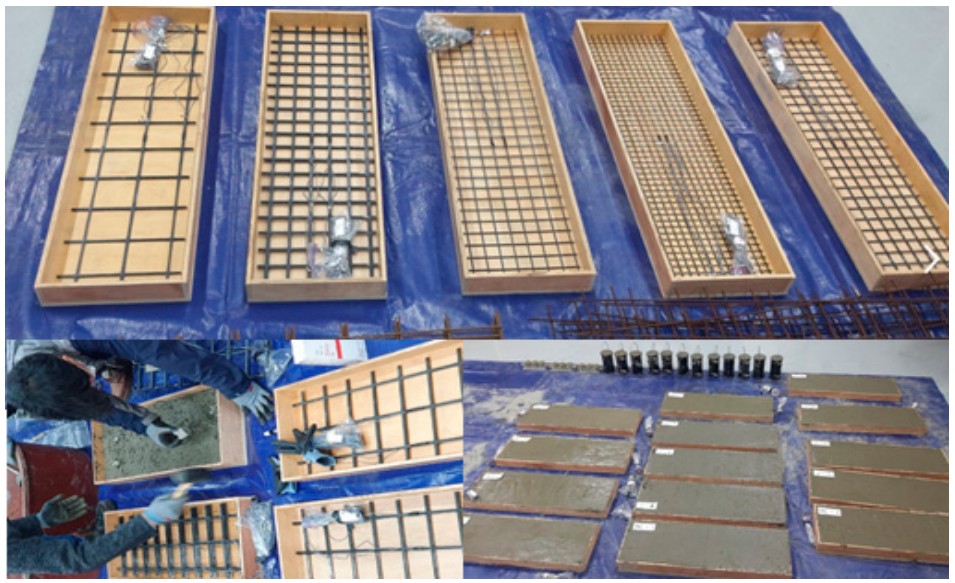

**Figure 4.** Manufacturing process for the specimens reinforced with carbon grids.

Table 4 summarizes the calculated crack and flexural capacities of the specimens based on the equilibrium of forces and strain compatibility. Tensile failure in the steel-reinforced specimen was expected to be caused by the yielding of steel, whereas compression failure at the compression edge of a section in the carbon grid-reinforced specimens was expected to be due to mortar crushing. Among the carbon grid-reinforced specimens, specimen J5 with carbon grid CR16 exhibited a tensile force approximately five times higher than that of the steel-reinforced specimen; nevertheless, their flexural strengths were similar.

**Table 4.** Crack and flexural strength results for the specimens.

| Specimen | Crack Strength | | Flexural Strength | | Failure Mode |
|---|---|---|---|---|---|
| | Moment (kNm) | Load (kN) | Moment (kNm) | Load (kN) | |
| G1 | 0.38 | 1.67 | 2.23 | 9.89 | |
| G2 | 0.38 | 1.67 | 2.17 | 9.65 | |
| G3 | 0.38 | 1.67 | 1.69 | 7.49 | |
| J1 | 0.38 | 1.67 | 1.90 | 8.45 | |
| J2 | 0.38 | 1.67 | 2.32 | 10.33 | Mortar crushing |
| J3 | 0.38 | 1.67 | 2.14 | 9.51 | |
| J4 | 0.38 | 1.67 | 2.84 | 12.62 | |
| J5 | 0.38 | 1.67 | 2.33 | 10.34 | |
| J6 | 0.38 | 1.67 | 2.60 | 11.54 | |
| RC | 0.38 | 1.67 | 2.97 | 13.2 | Steel yield |

### 2.2. Loading and Measurement Method

A three-point bending test was performed with a 1000 kN actuator (Figure 5) and a load applied to the center of the specimen with a displacement control of 5 mm/min.

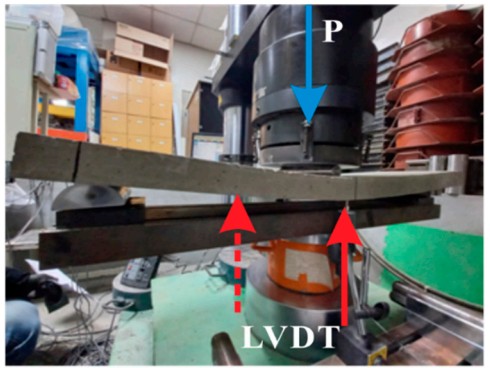

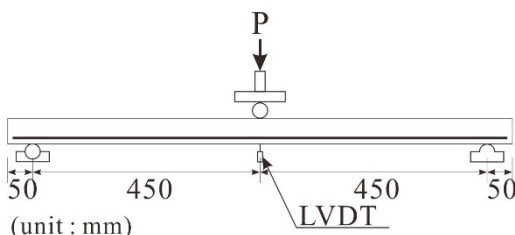

**Figure 5.** Test set-up.

The load was measured using a load cell embedded in the actuator. The mid-span deflection of the specimen was measured by installing two CDP 1000 linear variable differential transformers (LVDTs) (Tokyo measuring instruments lab) under both ends of the central section (Figure 5).

Giese et al. [13] reported that the strain of textile reinforcement could be measured using strain gauges; in this paper, four strain gauges were installed on the center of the two longitudinal strands located at the center of the cross-section for specimens G1 to G3, and the center and end of the cross-section for specimens J1 to J6, to measure the strain of the carbon grids (Figure 2).

The strain of the steel-reinforced specimen was measured by installing two strain gauges on the steel located at the center of the cross-section of the specimen.

## 3. Results and Discussion

### 3.1. Crack and Failure Geometry

Figure 6 presents the final failure geometry of the carbon grid-reinforced specimens.

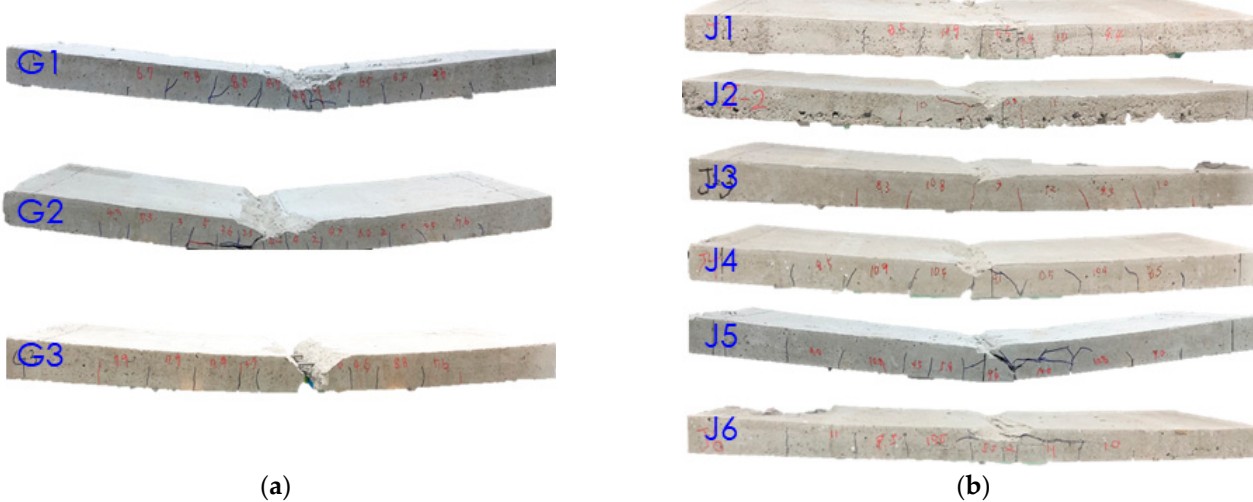

(**a**)　　　　　　　　　　　　　　　　　　　　　　　(**b**)

**Figure 6.** Failure of the specimens reinforced with: (**a**) biaxial warp-knitted; and (**b**) cross-laminate structured carbon grids.

The specimens with both biaxial warp-knitted and cross-laminate structured carbon grids demonstrated the occurrence of cracks at the center of each specimen. Subsequently, new cracks occurred from the center of the specimen to both ends owing to the applied load. At the center of the specimen, the mortar was crushed and separated.

In specimens G1, G2, and G3 reinforced with the biaxial warp-knitted structured carbon grids, 11, 17, and 11 cracks occurred, respectively (Table 5 and Figure 7). The

average crack spaces ranged from 45 to 63 mm, wider than the transverse strand spacings of the carbon grids. Furthermore, Figure 7 indicates that a bond crack between the carbon grid and mortar was observed in the case of specimen G2.

**Table 5.** Crack formation of the specimens.

| Specimen | Number of Cracks | Average Crack Space (mm) |
|----------|-----------------|--------------------------|
| G1 | 11 | 63 |
| G2 | 17 | 45 |
| G3 | 11 | 63 |
| J1 | 7 | 81 |
| J2 | 4 | 106 |
| J3 | 7 | 99 |
| J4 | 8 | 98 |
| J5 | 10 | 88 |
| J6 | 8 | 83 |

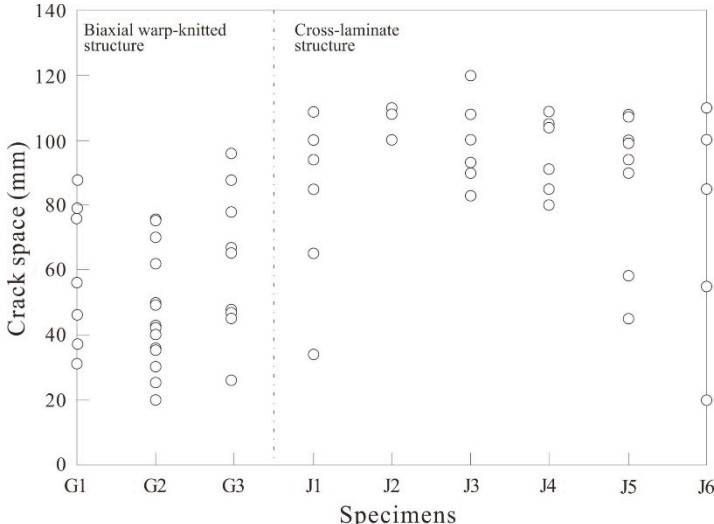

**Figure 7.** Specimen crack spaces.

Conversely, the specimens reinforced with the cross-laminate structured carbon grids had fewer cracks, ranging from four to ten, compared with those observed in the specimens reinforced with the biaxial warp-knitted structured carbon grids. Moreover, the average crack spaces ranged from 81 to 106 mm. The average crack space for specimen J2 was 106 mm, which was more than twice the transverse strand spacing of 50 mm. On the other hand, the average crack spaces of other specimens were smaller than the transverse strand spacing of 100 mm.

Bond cracks were also observed in specimen J2 (Figure 7); as in specimen G2, the average crack spaces in specimens J2 and G2 were considered to be more than twice the transverse strand spacings because of the poor bond between the carbon grids and mortar.

By the end of the experiment, no strand fractures were observed in the specimens reinforced with the cross-laminate structured carbon grids. Furthermore, Figure 8 indicates that only a few fibers of the strands were disconnected in specimen J5. However, in the case of specimens reinforced with the biaxial warp-knitted structured carbon grids, the longitudinal strands were fractured in specimens G3 and G1 when the mortar was removed from the center of the specimens after the experiment (Figure 8). Additionally, residual deformation of the carbon grid was observed in specimen G2. In the case of biaxial warp-knitted structured carbon grids, several voids occurred in the cross-section of the strands as they were manufactured using the hand lay-up method (Figure 3a). The strands appeared to be fractured owing to the increase in the applied force and the concentration of this force on the voids.

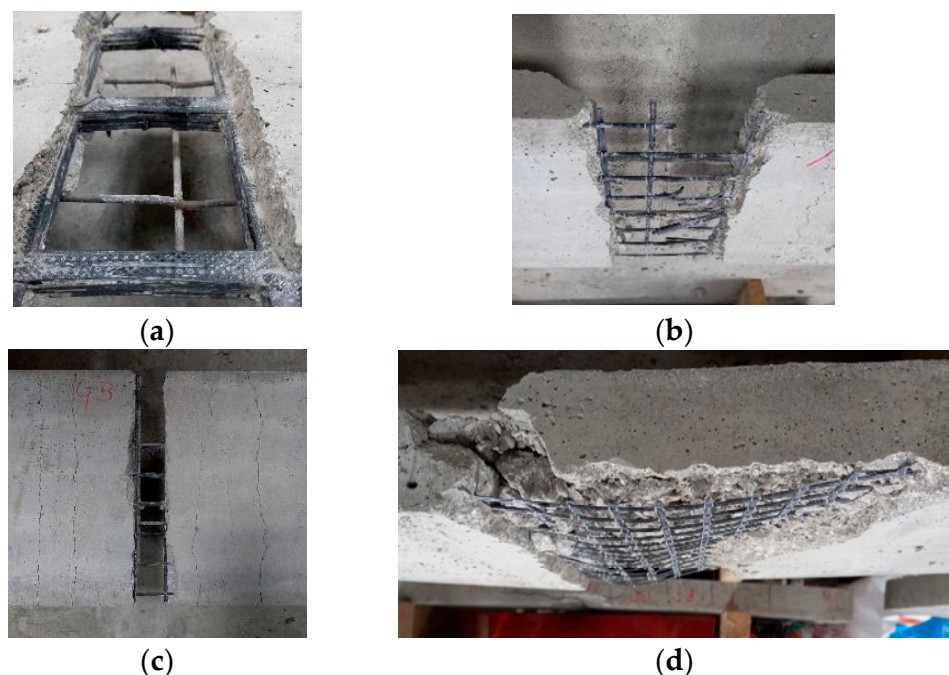

**Figure 8.** Damage observed in the carbon grids of specimens: (**a**) J5; (**b**) G1; (**c**) G3; and (**d**) G2.

### *3.2. Flexural Behaviors of Specimens with Carbon Grids*

3.2.1. Load–Deflection Relationship

In all specimens reinforced with the biaxial warp-knitted and cross-laminate structured carbon grids, cracks were initially observed at the center of each specimen. Furthermore, multiple cracks continuously developed toward the ends of the specimens, thereby significantly reducing the stiffness due to the transfer of tensile stress to the carbon grid (Figure 9). Subsequently, the maximum load was reached by mortar crushing in the mid-span of the specimen. Although the load in most specimens was reduced in stages after reaching the maximum load, specimens G2 and G3, which were reinforced with the biaxial warp-knitted structured carbon grids, exhibited a sharp load reduction.

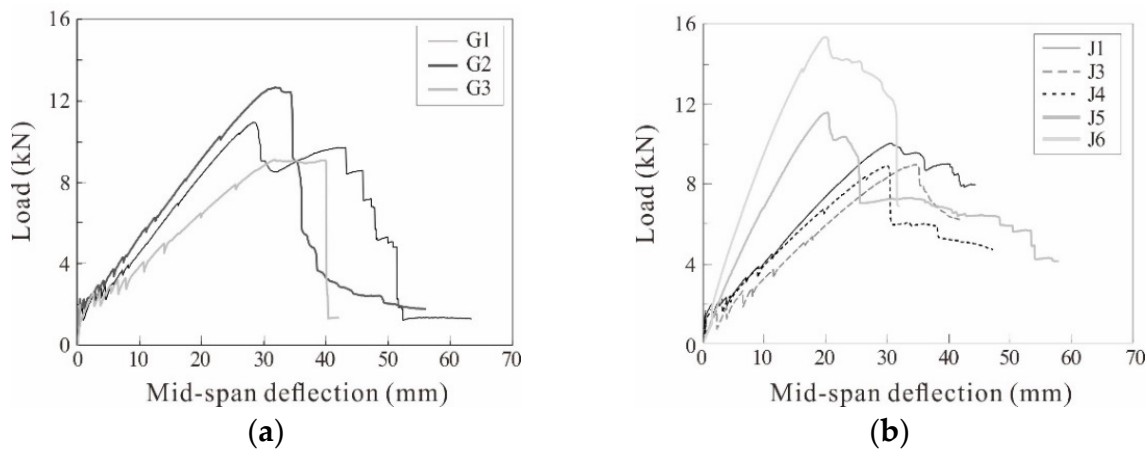

**Figure 9.** Relationship between the load and mid-span deflection for specimens reinforced with: (**a**) biaxial warp-knitted; and (**b**) cross-laminate structured carbon grids.

In the case of specimens reinforced with the biaxial warp-knitted structured carbon grids, specimen G2, with the highest tensile capacity, exhibited the largest crack formation and maximum loads (Table 6). The load of the specimens reinforced with the biaxial warp-knitted structured carbon grids decreased to approximately 11 to 14% of the maximum

load upon completion of the experiment owing to the fracture of the longitudinal strands and residual deformation of the carbon grid (Figure 9a).

**Table 6.** Test results.

| Specimen | Crack Formation | | Peak | |
|:---:|:---:|:---:|:---:|:---:|
| | Load (kN) | Deflection of Mid-Span (mm) | Load (kN) | Deflection of Mid-Span (mm) |
| G1 | 1.61 | 0.37 | 10.92 | 28.52 |
| G2 | 2.25 | 0.36 | 12.69 | 31.82 |
| G3 | 1.85 | 0.36 | 9.10 | 31.77 |
| J1 | 1.47 | 0.32 | 10.05 | 30.71 |
| J2 | 1.95 | 2.24 | 12.30 | 27.18 |
| J3 | 1.51 | 0.52 | 8.97 | 34.60 |
| J4 | 1.28 | 0.24 | 8.94 | 30.03 |
| J5 | 1.10 | 1.27 | 11.58 | 20.24 |
| J6 | 1.92 | 1.57 | 15.33 | 20.10 |
| RC | 1.00 | 1.18 | 14.73 | 14.95 |

In the case of specimens reinforced with the cross-laminate structured carbon grids, the crack formation load did not significantly differ depending on the type of carbon grid. Furthermore, the maximum load did not increase in proportion to the tensile force by the longitudinal strands (Table 6). In addition, specimen J2 had a smaller longitudinal strand spacing (50 mm) than specimen J3 with the 100 mm spacing, and exhibited an increase in stiffness after crack formation and maximum load due to the increase in the number of strands (Figure 10).

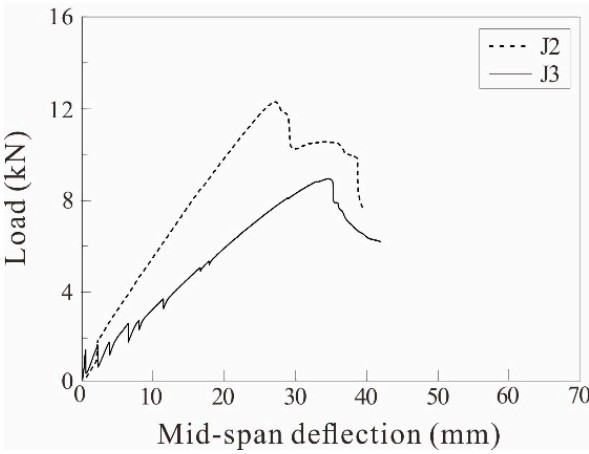

**Figure 10.** Relationship between the load and mid-span deflection for specimens considering grid longitudinal spacing.

### 3.2.2. Behavior of Crack Formation and Stabilization Stages

Table 7 summarizes the stiffness of the pre-cracking, crack formation, and crack stabilization stages, and Figure 11 illustrates the stiffness of the crack formation and stabilization stages. The stiffness of the pre-cracking stage indicates the stiffness between the beginning of the experiment and the initiation of crack formation. The stiffness of the crack formation stage represents the stiffness between the initiation of crack formation and the point at which cracks stop occurring continuously. The stiffness of the crack stabilization stage indicates the stiffness between the point at which cracks stop occurring continuously and the peak.

**Table 7.** Stiffness of the specimens at pre-cracking, crack formation, and crack stabilization stages.

| Specimen | Stiffness (N/mm²) | | | (2)/(1) | (3)/(1) |
|---|---|---|---|---|---|
| | Pre-Cracking [1] | Crack Formation [2] | Crack Stabilization [3] | | |
| G1 | 4296 | 325 | 334 | 8% | 8% |
| G2 | 6257 | 233 | 346 | 4% | 6% |
| G3 | 5149 | 193 | 242 | 4% | 5% |
| J1 | 4568 | 245 | 287 | 5% | 6% |
| J2 | 1146 | - | 415 | - | 53% |
| J3 | 2908 | 170 | 232 | 6% | 8% |
| J4 | 5465 | 268 | 255 | 5% | 5% |
| J5 | 1158 | - | 553 | - | 48% |
| J6 | 1918 | - | 817 | - | 89% |

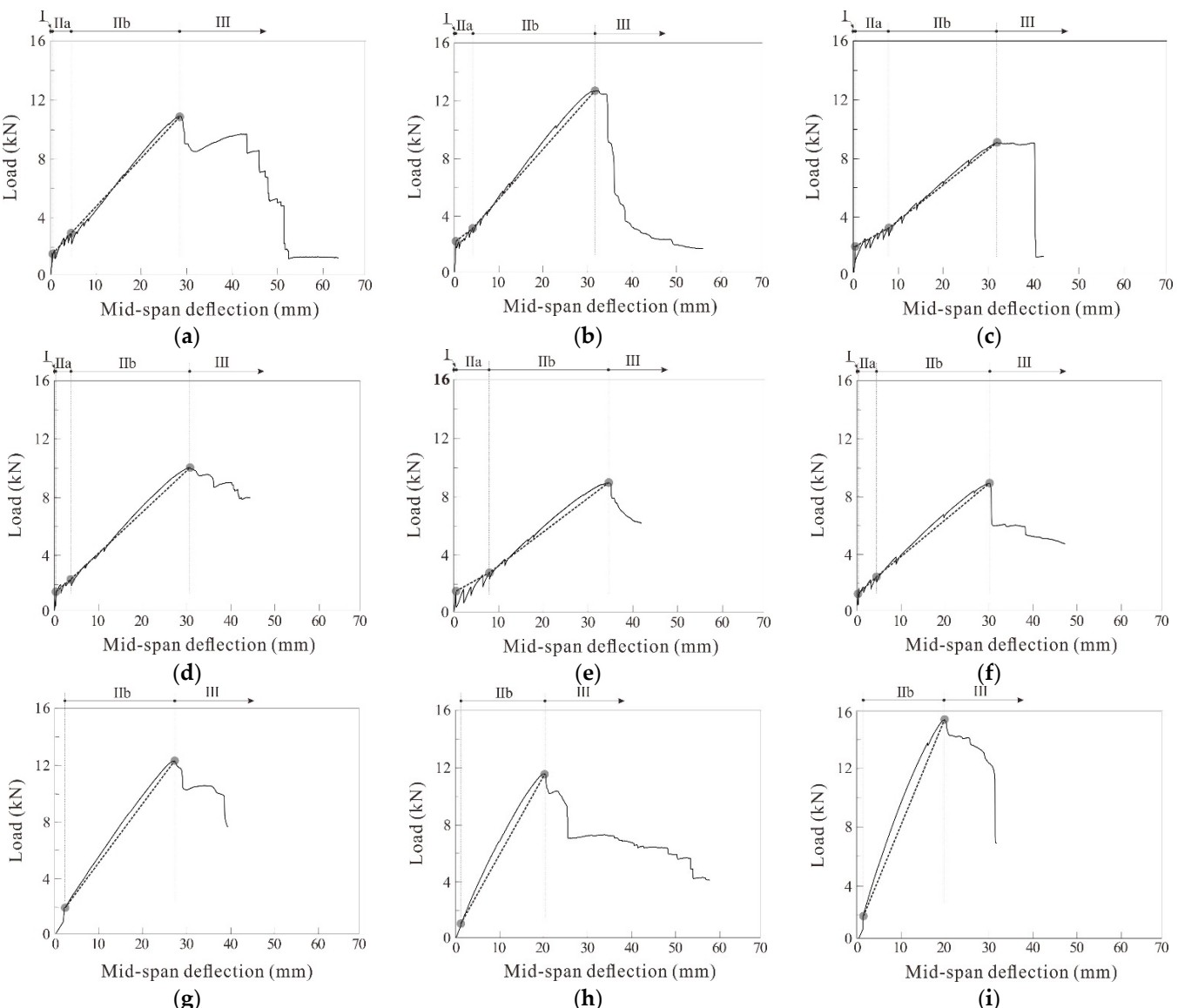

**Figure 11.** Stiffness in the crack formation and stabilization stages for the specimens: (**a**) G1; (**b**) G2; (**c**) G3; (**d**) J1; (**e**) J3; (**f**) J4; (**g**) J2; (**h**) J5; and (**i**) J6.

The stiffness of the crack formation and stabilization stages of the carbon grid-reinforced specimens ranged from 4 to 8% and 5 to 8% of the pre-cracking stiffness, respectively, indicating that the stiffness of the specimens was rapidly reduced by cracking, and the stiffness hardly recovered even after the stabilization of crack formation. Figure 11g–i indicate that among the specimens reinforced with the cross-laminate structured carbon grids, specimens J5 and J6 with the largest cross-sectional areas of the longitudinal strands and specimen J2 with a small strand spacing immediately exhibited the behavior of the crack stabilization stage without the crack formation, unlike the behavior observed in other specimens; in the case of these three specimens, the load significantly increased, and the mid-span deflection increased only marginally because of cracking. The stiffer carbon grids with larger cross-sectional areas or smaller spacing of the strands used in specimens J5, J6, and J2 might affect the behavior of crack formation stage.

The elastic modulus of the biaxial warp-knitted structured carbon grids was more than twice as high as that of the cross-laminate structured carbon grids. However, the stiffness of the specimens reinforced with the biaxial warp-knitted structured carbon grids in the crack formation and stabilization stages was not significantly different from that observed in the same stages of specimens J1, J3, and J4, which were reinforced with the cross-laminate structured carbon grids (Table 7). Moreover, the stiffness was lower than that observed in the crack stabilization stage of specimens J2, J5, and J6 because the stiffness of cracked section is associated with the effective moment of inertia [7], which is affected by both the total cross-sectional area and elastic modulus of the longitudinal strands. As shown in Figure 12, the stiffness of the specimens in the crack formation and stabilization stages tended to be proportional to the values obtained by multiplying the total cross-sectional area of the longitudinal strands by the elastic modulus of the longitudinal strands.

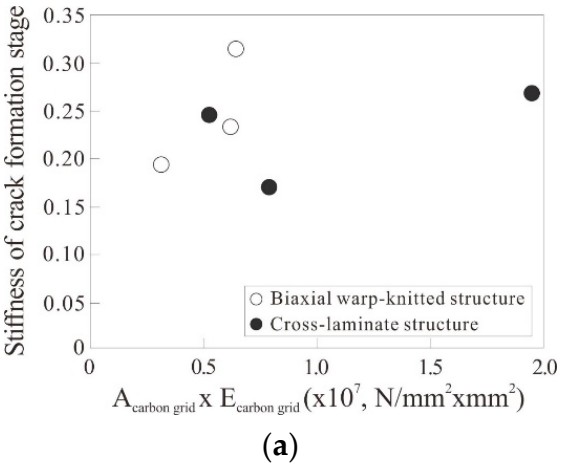 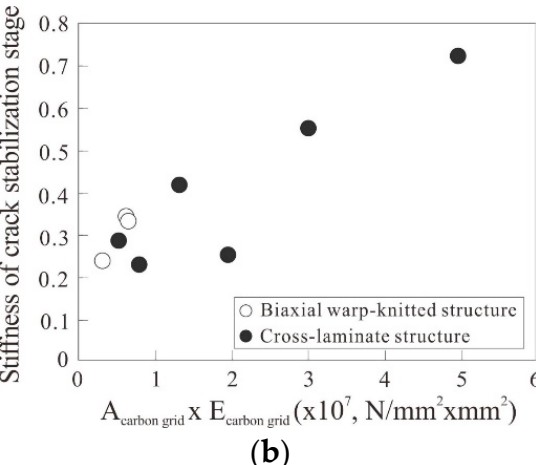

(**a**)  (**b**)

**Figure 12.** Comparison of stiffness: (**a**) crack formation; and (**b**) stabilization stages.

### 3.2.3. Post-Peak Behavior

Table 8 lists the ratios of strain to the tensile strain of the longitudinal strands in the carbon grids under and after the maximum load. Figure 13 illustrates the relationship between the mid-span deflection of the carbon grid-reinforced specimens and strain of the longitudinal strands; here, the strain indicates the value measured using the strain gauges installed on the strands (Figure 3).

Specimens G1, J1, J2, J3, J4, J5, and J6 exhibited a stage-wise load reduction tendency after reaching the maximum load (Figure 9) as the mid-span deflection of the specimens increased. The strains of the longitudinal strands under the maximum load ranged from 38 to 65% of the tensile strain (Table 7), indicating the occurrence of tensile stress that corresponded to 38 to 65% of the tensile strength.

**Table 8.** Ratios of strain to tensile strain.

| Specimen | At Peak | | After Peak | | |
|---|---|---|---|---|---|
| G1 | 0.64 | 0.80 | 0.83 | | |
| G2 | 0.64 | 0.70 | | | |
| G3 | 0.82 | 1.01 | | | |
| J1 | - | | | | |
| J2 | 0.54 | 0.55 | | | |
| J3 | 0.38 | 0.44 | | | |
| J4 | 0.53 | 0.56 | 0.55 | | |
| J5 | 0.38 | 0.47 | 0.58 | 0.70 | 0.93 |
| J6 | 0.65 | 0.83 | 1.02 | | |

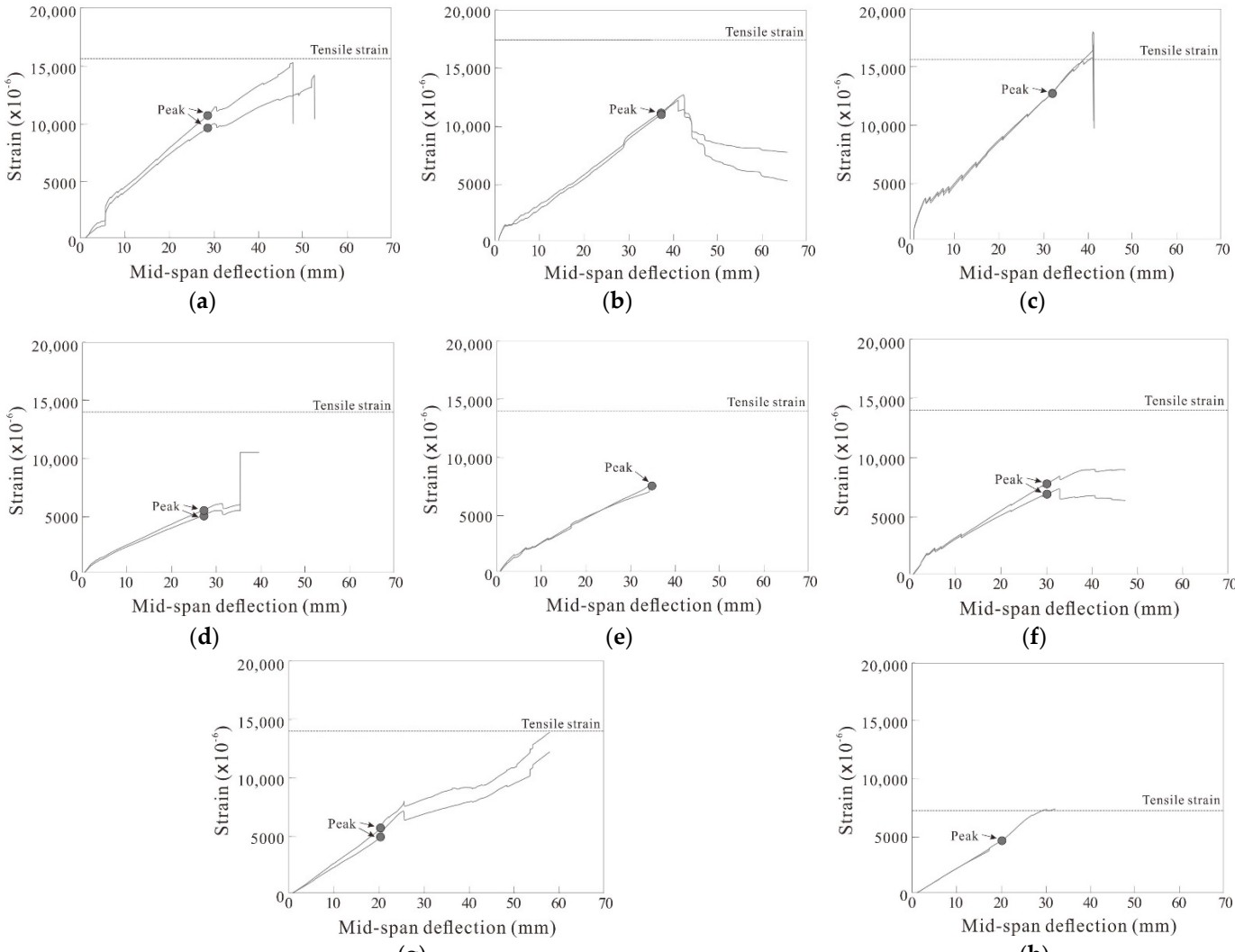

**Figure 13.** Relationship between the mid-span deflection and strain in the specimens: (**a**) G1; (**b**) G2; (**c**) G3; (**d**) J2; (**e**) J3; (**f**) J4; (**g**) J5; and (**h**) J6.

In the case of specimen G1, the load decreased after reaching the maximum load, and a certain amount of the load was recovered as the carbon grid resisted the load. However, the load rapidly decreased again as a strain of approximately 80% of the tensile strain was measured. Finally, the load sharply decreased and failure occurring as a strain of approximately 83% of the tensile strain was measured. In the case of specimen J5, a certain amount of the load was recovered as the carbon grid resisted the load after reaching the maximum load; nevertheless, the load rapidly decreased again as a strain that corresponded

to approximately 47% of the tensile strain was measured. The load gradually decreased, and the final failure occurred because of the sharp reduction in load when a strain of approximately 93% of the tensile strain was measured. In the case of specimen J6, the load rapidly decreased after reaching the maximum load as a strain corresponding to approximately 83% of the tensile strain was measured. Subsequently, the load sharply decreased, and final failure was reached as a strain of approximately 102% of the tensile strain was observed.

In the case of specimen G1, J5 and J6, a difference was observed between the strain measurement results and the damage status of the longitudinal strands. The strain gauges were practically installed on the resin on the strand surface and recorded the deformation of resin and some fibers only at the position where the strain gauge was attached; thereby, the measured values could be affected by the strand surface condition. The biaxial warp-knitted structured carbon grid used in specimen G1 used epoxy with relatively high physical properties as resin and had the small cross-sectional area of the strands and the cross-sectional voids generated during manufacturing. Therefore, the strands were considered to fracture by the occurrence of strain corresponding to 83% of the tensile strain. Conversely, the cross-laminate structured carbon grid used in specimens J5 and J6 used vinyl ester, with lower physical properties than epoxy [23] and a larger cross-sectional area. Consequently, it was considered that the strands were not fractured despite the occurrence of strain that corresponded to approximately 93 and 102% of the tensile strain.

The load in specimens G2 and G3, with the biaxial warp-knitted structured carbon grids, tended to reduce sharply after reaching the maximum load (Figure 9). In the case of specimen G2, a strain corresponding to approximately 64% of the tensile strain was measured under the maximum load. However, the load rapidly decreased after reaching the maximum load, and final failure occurred owing to the expansion of the bond crack generated between the carbon grid and mortar, unlike the behavior observed in specimens G1, J1, J2, J3, J4, J5, and J6. In other words, it was considered that the load in specimen G2 rapidly decreased immediately right after reaching the maximum load, although approximately 64% of the tensile stress was applied to the longitudinal strands under the maximum load because of the poor bond between the carbon grid and mortar. The strands were not fractured, but the residual deformation on the carbon grid occurred at the end of the experiment (Figure 8).

Among the specimens, the largest tensile stress was observed in specimen G3 at a strain corresponding to approximately 82% of the tensile strain was measured when the maximum load was reached. Although a certain amount of the load was maintained after reaching the maximum load, the load suddenly dropped and final failure occurred as the cracks at the center of the specimen expanded and the longitudinal strands were fractured under the application of a load higher than the tensile strength because of the brittle fracture characteristics of CFRP composites [24].

### 3.2.4. Idealized Load and Mid-Span Deflection of Specimens with Carbon Grids

Based on the experimental results, Figure 14 illustrates the idealized relationship between the load and mid-span deflection of the carbon grid-reinforced specimens.

The experimental results indicate that after cracking, the stiffness decreased to approximately 8% or lower than the stiffness before cracking, except for the three specimens reinforced with relatively stiffer carbon grids. Additionally, the maximum load was reached when the stiffness was only marginally recovered, even at the crack stabilization stage. In other words, the load gradually increased owing to the increase in the mid-span deflection with a similar stiffness at the crack formation and stabilization stages even though the stiffness decreased sharply because of the crack occurrence, unlike the behavior observed in flexural members that used textile reinforcement without resin (Figure 2). Therefore, the load and mid-span deflection relationship might be modeled as a bilinear curve in which the stiffness changes around the crack initiation point (Figure 14).

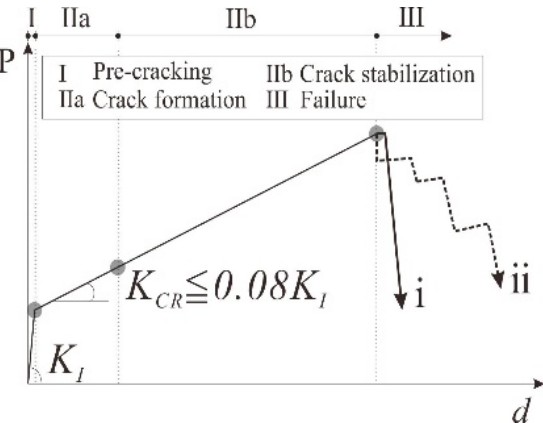

**Figure 14.** Idealized load and mid-span deflection relationship.

In accordance with the experimental results, when the strain of the longitudinal strands was less than approximately 80% of the tensile strain under the maximum load, the load was found to decrease in stages as the load resisted by the mortar was transferred to the carbon grid when the bond between the carbon grids and mortar was secured (ii in Figure 14). Conversely, it was found that the load resisted by the mortar could not be transferred to the carbon grid after reaching the maximum load, while the load decreased rapidly when the bond between the mortar and carbon grid was not secured, or strain corresponding to approximately 80% of the tensile strain or higher occurred on the longitudinal strands under the maximum load (i in Figure 14).

*3.3. Comparison between the Experimental and Calculated Results*

The maximum load did not increase as the tensile capacity by the longitudinal strands of the carbon grid-reinforced specimens increased (Figure 15). Particularly, in the case of specimens reinforced with the cross-laminate structured carbon grids, the maximum load did not increase in proportion to the tensile capacity by the longitudinal strands because all specimens reached the maximum load by mortar crushing. As shown in Table 7, the strains of the longitudinal strands under the maximum load ranged from 38 to 82% of the tensile strain. Particularly, the longitudinal strands of the cross-laminate structured carbon grids exhibited a smaller strain than the longitudinal strands of the biaxial warp-knitted structured carbon grids. On the other hand, the maximum load tended to be higher in the specimens reinforced with small cross-sectional areas and spacings of the longitudinal strands but high tensile strength compared with the specimens reinforced with large cross-sectional areas and spacings of the longitudinal strands but low tensile strength.

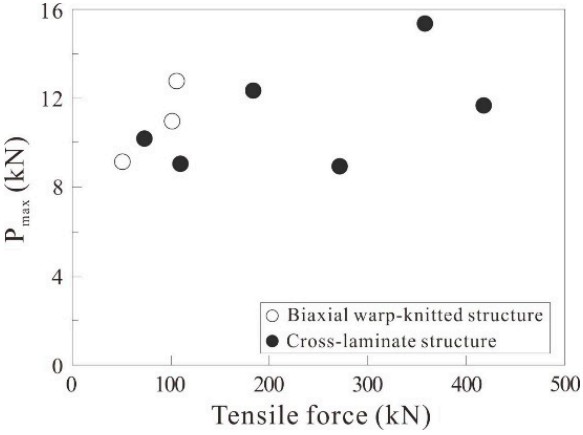

**Figure 15.** Relationship between carbon grid tensile force and maximum load.



Figure 16 presents the crack and flexural strength ratio of experimental and calculation results, considering the tensile force by the longitudinal strands.

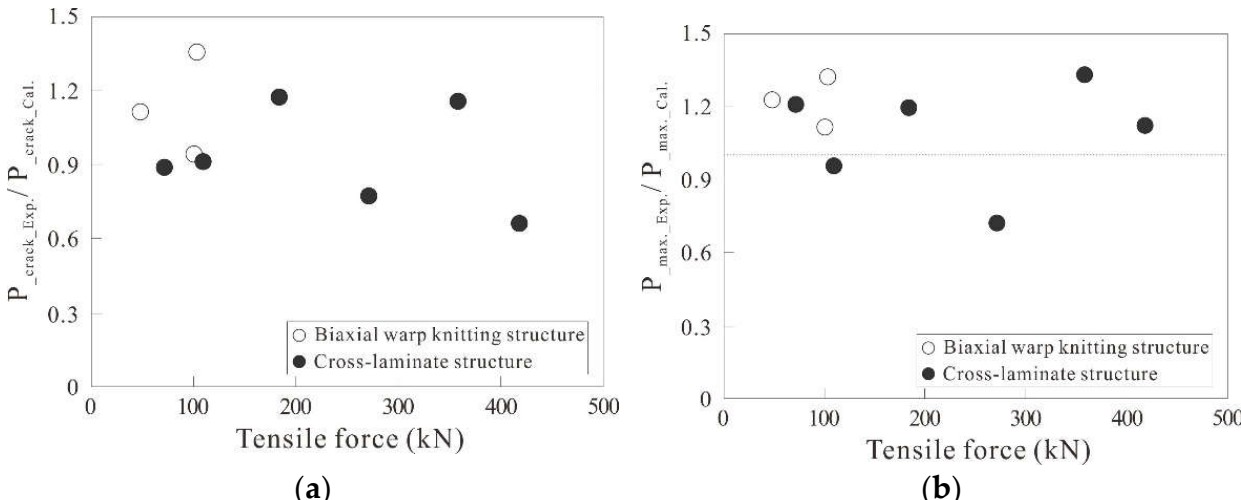

**Figure 16.** Strength ratio of experimental and calculation results according to carbon grid tensile force: (**a**) crack; and (**b**) flexural strengths.

The crack strength ratio ranged from 0.77 to 1.35, and the flexural strength ratio ranged from 0.71 to 1.33. The maximum load was generally higher than the flexural capacity. Particularly, in the case of specimens reinforced with the biaxial warp-knitted structured carbon grids with a relatively low tensile capacity, the maximum load was more than 1.0 times higher than the flexural capacity. In the case of specimen G2, which exhibited a poor bond between the carbon grid and mortar, the maximum load was 1.3 times higher than the flexural capacity, indicating that the flexural capacity was conservatively evaluated.

Flexural stiffness is usually calculated using the moment of inertia related to the section states: the gross moment of inertia, $I_g$, for the uncracked section and effective moment of inertia, $I_e$, for the cracked section. It is known that in the case of TRC members, the effective moment of inertia becomes lower compared with that of RC members because of the different material properties of FRP reinforcement. Therefore, researchers [7,25] proposed the formation of an effective moment of inertia which modified Branson's equation for RC members [26] based on experimental results using glass fiber-reinforced plastic (GFRP) bars, as presented in Equation (1).

$$I_e \frac{I_{cr}}{1 - \left(\frac{M_{cr}}{M_a}\right)^2 \left[1 - \frac{I_{cr}}{I_g}\right]} \tag{1}$$

where, $I_{cr}$ is the cracked transformed moment of inertia (mm$^4$) of the following Equation (2), $M_{cr}$ is the cracking moment (Nmm), and $M_a$ is the maximum load moment for the deflection calculation (Nmm).

$$I_{cr} = \frac{bd^3}{3}k^3 + nA_{CFRP}d^2(1-k)^2 \tag{2}$$

where, $b$ is the width (mm), $d$ is the effective depth (mm), $n$ is the elastic modulus ratio of the carbon grid and concrete, $A_{CFRP}$ is the longitudinal strand area (mm$^2$), and $k$ is the normalized neutral axis depth of the cracked section.

In all specimens, the calculated stiffness of the cracked sections using $I_e$ in Equation (1) was revealed to be 1.5 times higher than the stiffness of the crack formation stage from the experimental results (Table 9 and Figure 17). Here, the calculated crack and flexural moments in Table 4 were used for $M_{cr}$ and $M_a$, respectively. Meanwhile, the calculation (Table 9) showed that $I_e$ and $I_{cr}$ were similar, and the calculated stiffness of the cracked

sections using $I_{cr}$ in Equation (2) was also shown to be 1.2 times higher than the stiffness of the crack formation stage from the experimental results. Equation (1) was proposed based on GFRP bars with an elastic modulus of 40 GPa and tensile strength of 690 MPa as reinforcement. These differences in physical properties between carbon grids and the GFRP bars are considered to affect the calculation results; the calculation values are higher than the experimental results because of the higher elastic moduli and tensile strengths of the carbon grids.

**Table 9.** Stiffness of cracked sections by effective and cracked transformed moment of inertia.

| Specimen | $I_{cr}$ (mm$^4$) | $I_e$ (mm$^4$) | Stiffness of Cracked Section (N/mm) | |
|---|---|---|---|---|
| | | | Based on $I_{cr}$ | Based on $I_e$ |
| G1 | 287,945 | 295,566 | 538 | 524 |
| G2 | 278,931 | 286,721 | 522 | 508 |
| G3 | 151,084 | 158,556 | 289 | 275 |
| J1 | 217,652 | 225,827 | 411 | 396 |
| J2 | 437,175 | 447,188 | 814 | 796 |
| J3 | 296,093 | 304,562 | 554 | 539 |
| J4 | 600,818 | 609,404 | 1109 | 1094 |
| J5 | 484,276 | 495,148 | 901 | 881 |
| J6 | 676,201 | 687,438 | 1251 | 1231 |

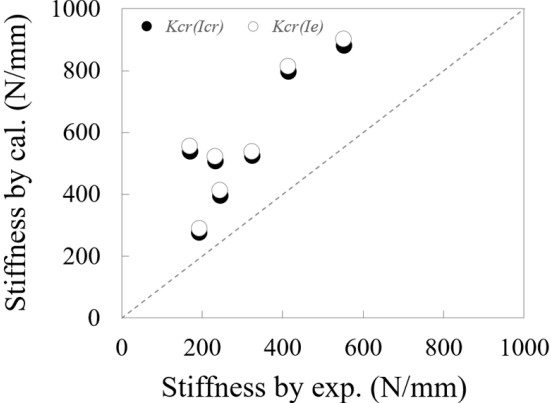

**Figure 17.** Stiffness ratios for experimental and calculation results.

### 3.4. Comparison of Flexural Behaviors with RC

The tensile capacity of the carbon grids in the carbon grid-reinforced specimens was 0.6 to 5.0 times higher than that of the steel in specimen RC (Table 1); however, the maximum load of the carbon grid-reinforced specimens ranged from 60 to 110% of the maximum load of the steel specimen RC (Figure 18), because specimen RC reached its maximum load owing to the yielding of steel, whereas the carbon grid-reinforced specimens reached their maximum load because of mortar crushing. In this case, only tensile stress corresponding to 38–82% of the tensile strength was observed on the longitudinal strands, as indicated in Table 7. In the case of specimens J4 and J5 reinforced with cross-laminate structured carbon grids FTG-CR13 and FTG-CR16, which were calculated to have a flexural capacity similar to that of specimen RC, the maximum load ranged from 60 to 79% of the maximum load of specimen RC.

The stiffness of specimens J4 and J5 was significantly lower than that of specimen RC after cracking (Figure 19) because the elastic modulus of the longitudinal strands was 0.5 times lower than that of steel, whereas the total cross-sectional areas of the strands in the carbon grids arranged in specimens J4 and J5 were similar to or larger than that of steel by 0.91 and 1.40 times, respectively. Consequently, the mid-span deflections of specimens J4 and J5 under the maximum load were 2.00 and 1.35 times higher than that of specimen

RC, respectively (Table 5). Furthermore, in the case of specimen RC, the load gradually decreased even after reaching the maximum load owing to the ductile behavior after the yielding of steel; approximately 81% of the maximum load was maintained at the end of the experiment. However, in the case of specimens J4 and J5, the load sharply decreased to approximately 67 and 61% of the maximum load, respectively, as the load resisted by the mortar was transferred to the carbon grids after reaching the maximum load due to the heterogeneous bond between the mortar and carbon grids [27]. The load finally decreased to approximately 53 and 36% of the maximum load for specimens J4 and J5, respectively, without carbon grid fracture at the end of the experiment.

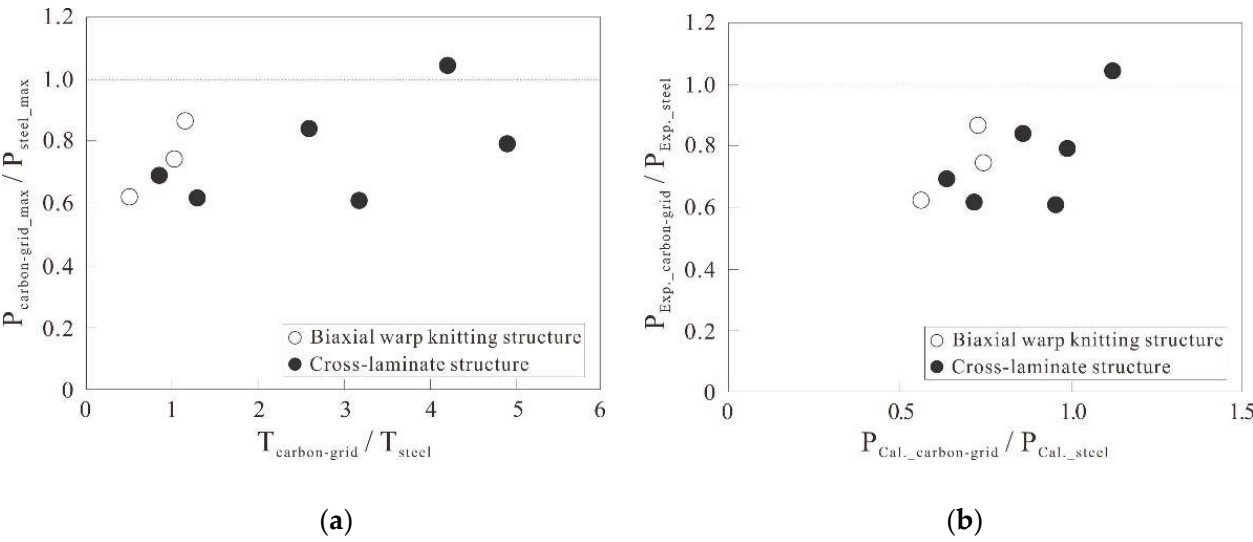

(**a**) (**b**)

**Figure 18.** Comparison of the maximum load between the specimens reinforced with carbon grids and steel, based on: (**a**) tensile force; and (**b**) flexural strength.

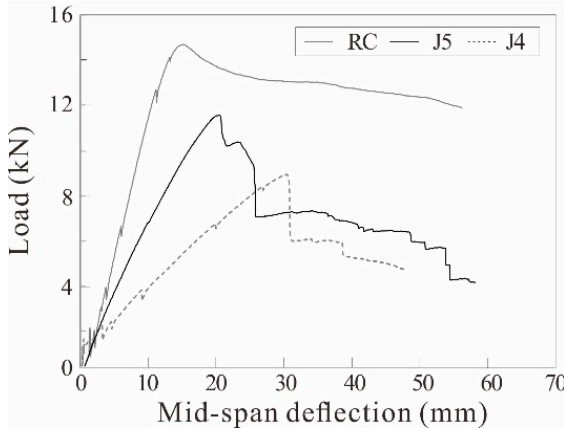

**Figure 19.** Flexural behavior of the specimens reinforced with carbon grids and steel.

## 4. Conclusions

In this study, the flexural behavior of TRC one-way slabs was evaluated by performing a three-point bending test on nine carbon grid-reinforced specimens and a steel-reinforced specimen. Carbon grids manufactured by two different methods with varying cross-sectional areas, spacings, tensile strengths, and elastic moduli of longitudinal strands were incorporated into the specimens. The study findings can be summarized as follows.

1.  The specimens reinforced with the biaxial warp-knitted structured carbon grids exhibited smaller crack spacings and more cracks than the specimens reinforced with the cross-laminate structured carbon grids. This was considered to be an effect of the

      smaller strand spacing of the biaxial warp-knitted structured carbon grid than that of the cross-laminated structured carbon grid.

2.    The biaxial warp-knitted structured carbon grids had smaller cross-sectional areas and spacings of strands but a higher tensile strength than the cross-laminate structured carbon grids. Consequently, the specimens reinforced with the biaxial warp-knitted structured carbon grids had a lower tensile capacity than the specimens reinforced with the cross-laminated carbon grids. However, the maximum load tended to be higher in the specimens reinforced with the biaxial warp-knitted structured carbon grids than in the specimens reinforced with the cross-laminate structured carbon grids, even though the tensile capacity was similar or low. Specimen G2, which had the highest tensile capacity among the specimens reinforced with the biaxial warp-knitted structured carbon grids, also exhibited a maximum load 1.10 times higher than specimen J5 which had the highest tensile capacity among all carbon grid-reinforced specimens.

3.    All specimens reached the maximum load simultaneously with mortar crushing, and little damage occurred to the cross-laminated structured carbon grids by the end of the experiment. In particular, the specimens reinforced with the cross-laminated structured carbon grids seem to have used excessive carbon grids compared with the specimen conditions such as the effective depth and compressive strength of mortar.

4.    When the maximum load was reached, strains corresponding to 38–82% of the tensile strain were observed on the longitudinal strands of the carbon grids. The load then decreased as the load resisted by the mortar was transferred to the carbon grids immediately after the peak. The flexural strengths calculated based on the force equilibrium and strain compatibility equations were also found to conservatively evaluate the maximum loads revealed by the experiment.

5.    The load decreased immediately after the maximum load, but the load reduction stopped, or the load slightly increased, with the increasing mid-span deflection in the case of the specimens in which a stress of less than 80% of the tensile strength was applied to the longitudinal strands under the maximum load. The experimental results validate (i) the design for generating tensile stress in a carbon grid that corresponds to less than 80% of the tensile strength at the ultimate state and (ii) the requirement for a securing bond between a carbon grid and mortar to prevent brittle failures right after reaching the maximum load caused by the carbon-grid fracture.

6.    Among the specimens reinforced with the cross-laminate structured carbon grids, the specimens reinforced with the stiffer carbon grids because of the largest cross-sectional areas or small spacing of strands immediately exhibited the behavior of the crack stabilization stage without crack formation.

7.    Except for the stiffer carbon grid-reinforced specimens, the stiffness of the carbon grid-reinforced specimens after cracking decreased to approximately 8% or lower than those before cracking. Moreover, the stiffness did not significantly increase in the crack stabilization stage compared to that observed in the crack formation stage, thereby the load and mid-span deflection relationship might be modeled as a bilinear curve where the stiffness changes around the crack initiation point. Meanwhile, the calculated stiffness of the carbon grid-reinforced specimens after cracking, based on the existing equation of the effective moment of inertia for TRC members, was revealed to be 1.5 times higher than the experimental results; this was because the equation was proposed based on TRC members reinforced with GFRP bars, which had a lower tensile modulus and strength than the carbon grids in this paper.

8.    Although the carbon grids with a higher tensile force than steel were arranged in the specimens, their maximum load ranged from approximately 60 to 110% of the maximum load of the steel-reinforced specimen. Additionally, the stiffness of the carbon grid-reinforced specimens after cracking was significantly lower than that in the steel-reinforced specimen. Although the steel-reinforced specimen exhibited ductile behavior, the carbon grid-reinforced specimens exhibited a brittle behavior,

wherein the load sharply decreased in stages as the load resisted by the mortar was transferred to the carbon grids.

9. The test results were drawn from nine one-way slab specimens with two different types of carbon grids, and mortar with one level of compressive strength. Consequently, further research considering various levels of compressive strength, the effect of transverses strands, the prediction of cracked section stiffness, and so on are required.

**Author Contributions:** Conceptualization, K.-M.K. and J.-H.C.; conducting experiments, J.-H.C.; analyzing data and writing the manuscript, K.-M.K. All authors have read and agreed to the published version of the manuscript.

**Funding:** This work was supported by a grant from the Korea Agency for Infrastructure Technology Advancement (KAIA), funded by the Ministry of Land, Infrastructure and Transport (Grant 22CFRP-C163400-02).

**Institutional Review Board Statement:** Not applicable.

**Informed Consent Statement:** Not applicable.

**Data Availability Statement:** Not applicable.

**Conflicts of Interest:** The authors declare no conflict of interest.

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
