# Peer review of "Flexural Behavior of One-Way Slab Reinforced with Grid-Type Carbon Fiber Reinforced Plastics of Various Geometric and Physical Properties"

_applsci, doi:10.3390/app122312491_

Round 1
Reviewer 1 Report (New Reviewer)
The authors studied the behaviour of Textile reinforced concrete. They mentioned that there is limited experimental research on the flexural behavior of these members. I suggest that you add a subtitle of Research significance to mention what is new in this research and different from what is well known in the literature.
Author Response
1. We appreciate the comment. We changed the title to “Flexural behavior of one-way slab reinforced with grid-type carbon fiber reinforced plastics of various geometric and physical properties” in order to present the difference from other researches.

Reviewer 2 Report (New Reviewer)
In this study, the flexural behavior of TRC one-way slabs was evaluated by performing a three-point bending test on nine specimens reinforced with carbon grids and a specimen reinforced with steel. Carbon grids manufactured by two different methods with varying cross-sectional areas, spacings, tensile strengths, and elastic moduli of longitudinal strands were incorporated into the specimens. The paper is well written and the topic is of interest to the readers. The following comments are advised to enhance this paper.
1. Fiber textile is different from FRP grid. In the introduction section, this should be mentioned and discussed.
2. A series of recent studies on FRP grid reinforced concrete plates, which are of high relevance to this study, (e.g., Eng Struct 2022 272 115020) should be properly cited and commented.
3. How was the concrete cover thickness controlled during concrete casting?
4. The authors were testing concrete slabs with CFRP grid reinforcement. The authors are advised to explain the effects of fibers in the transverse direction.
5. crack spaces ranged from 81 to 106 mm: this should be caused by concrete separation, the crack width should not be as wide as the spacing.
6. The authors are advised to provide design recommendations for FRP grid reinforced concrete slabs, As the crack strength is small and the FRP capacity is sufficient, can the slabs be designed with cracks? What should be the design ultimate state for the FRP grid reinforced concrete slabs?
Author Response
1. We appreciate the comment, and fully agreed with your opinion. The grid-type FRPs are manufactured by impregnating them with resin differing from textiles that are manufactured by being woven in grid forms. The members reinforced with grid-type FRPs have been also reported to exhibit the improved tensile behavior compared with the members reinforced with textiles. These kinds of things were mentioned in the introduction section as follows;
“Among them, the grid-type FRP reinforcement is a type of textile-reinforced concrete (TRC) structure [14] that uses fibers woven in grid form as the reinforcement for concrete. Typical textile reinforcement is produced by weaving only fibers, such as glass, aramid, carbon, and basalt fibers, in grid form without impregnating them with resin. Although the grid-type FRP reinforcement exhibits higher tensile strength than reinforcing bars, the absence of plastic deformation capacity results in rapid brittle fracture after reaching the tensile strength [7]. A few researchers [15, 16] have reported that when fibers are impregnated or coated with resin as observed in FRP reinforcement, the tensile behavior of TRC members is improved in terms of smaller crack gaps, enhanced stable crack geometry, and increased maximum load.”
2. We appreciate the comment. The reference what you mentioned has been added at the proper locations.
- Zeng, J. J.; Zeng, W. B.; Ye, Y. Y.; Liao, J. J.; Zhuge, Y.; Fan, T. H. Flexural behavior of FRP grid reinforced ultra-high-performance concrete composite plates with different types of fibers, Eng. Struct. 2022, 272, 115020.
3. The carbon grids, especially the biaxial warp knitting structured carbon grids had the small strand cross-sectional areas and could be bent by pouring mortar. Therefore, in the case of carbon grid-reinforced specimens, the mortar was initially poured to the height corresponding to the cover thickness in order to secure the cover thickness.
4. We appreciate the comment. We had identified some effect of transverse strands on the tensile behavior through other tensile tests, and in this paper, the necessity of a further study on the transverse strand effect was added in the conclusion section as follows;
“The test results were drawn by nine one-way slab specimens with two different types of carbon grids and mortar with one level of compressive strength. Consequently, further researches considering various levels of compressive strength, the effect of transverses strands, the prediction of cracked section stiffness and so on are considered to be required.”
5. We appreciate the comment. In the case of specimens reinforced with the cross-laminate structured carbon grids, the average crack spaces ranged from 81 to 106 mm. Among the specimens reinforced with the cross-laminate structured carbon grids, the carbon grid with the strand spacing of 50 mm was used only for specimen J2, and the carbon grids with the strand spacing of 100 mm were used for other specimens. In the case of specimen J2, the average crack space was 106 mm which was more than twice of the strand spacing, the average crack spaces below 99 mm, that was smaller than the strand spacing were occurred on other specimens. In the case of specimen J2, a bond crack was observed, and the average crack space was considered to get larger than the strand spacing because of the poor bond between the carbon grid and mortar. These kinds of things revised and added in section 3.1 as follows;
“ Moreover, the average crack spaces ranged from 81 to 106 mm. The average crack space of specimen J2 was 106 mm, and it was more than twice of the transverse strand spacing of 50 mm. On the other hand, the average crack spaces of other specimens were smaller than the transverse strand spacing of 100 mm.
Bond cracks were also observed on specimen J2 (Figure 7) as in specimen G2, the average crack spaces of specimens J2 and G2 were considered to occur more than twice the transverse strand spacings because of the poor bond between the carbon grids and mortar.”
6. We appreciate the comment. The absence of plastic deformation capacity in carbon fiber reinforced polymer (CFRP) results in rapid brittle fracture of the members reinforced with grid-type CFRP reinforcement after reaching the tensile strength. Therefore, we were proposed that a design for generating tensile stress of grid-type CFRP reinforcement that corresponds to less than 80% of the tensile strength at the ultimate state and securing bond between grid-type CFRP reinforcement and mortar was required to prevent the brittle failures of the members reinforced with grid-type CFRP reinforcement in the conclusion section as follows;
“The load decreased right after the maximum load, but the load reduction was stopped or the load slightly increased with the increasing mid-span deflection in the case of the specimens in which a stress of less than 80% of the tensile strength was applied to the longitudinal strands under the maximum load. The experimental results validate that a design for generating tensile stress of a carbon gird that corresponds to less than 80% of the tensile strength at the ultimate state and securing bond between a carbon grid and mortar is required to prevent brittle failures right after reaching the maximum load caused by the fracture of a carbon grid.”

Reviewer 3 Report (New Reviewer)
The manuscript reports an experimental study on the flexural behavior of a one-way slab reinforced with grid-type CFRPs. The topic is interesting and the subject falls into the scope of the journal. However, the presentation of the manuscript should be significantly improved before any consideration for publication. The following points are provided for enhancing the manuscript's quality.
It's better not to use abbreviations in the manuscript title. In the abstract, write the abbreviated words, such as TRC and CFRP, even if they are very common.
The abstract is not clear to the reader regarding the samples used and their outcomes. More attention has to be given here, I recommend rewriting.
What is the scale of your slab? Why did you have to choose such a size for the experiment? An explanation is required. Include all the materials' properties in the Table.
How to design such a slab? The design procedure is needed.
The test setup is not clear? Provide a full sketch and which standard you use, it's better to write a test setup supported by the detailed sketch and testing protocol. A graphical illustration is needed for the test setup with the given photos. The fact is that the reader is lost to understand that you have 3 ( as shown in figure 2 ) or 5 ( as shown in figure 4 ) arrangement samples
The grammatical and language errors are significant. Need extensive language editing.
Conclusions are not related at all. In the manuscript, especially in the conclusion, please be more specific, provide more details and discussions
Author Response
1. We appreciate the comment. We changed the title without using abbreviations to the below;
“Flexural behavior of one-way slab reinforced with grid-type carbon fiber reinforced plastics of various geometric and physical properties”
2. We appreciate the comment. We rewrote the abstract as below considering what you recommended.
“ Abstract: Textile reinforced concrete (TRC) structure has many advantages including corrosion resistance, but TRC is the novel composite material and the limited experimental researches on the flexural behavior of TRC members have been conducted. This paper aimed to experimentally evaluate the flexural behavior of TRC slabs reinforced with nine types of grid-type carbon fiber-reinforced polymer (CFRP) reinforcement (hereafter referred to as carbon grid) with varying cross-sectional areas, spacings, tensile strengths, and elastic moduli of longitudinal strands. The experimental results showed that the maximum load had a tendency to be higher in the specimens reinforced with the carbon grids of small cross-sectional areas and spacings of strands but the high tensile strength. The cross-sectional area and spacing were also revealed to influence on the crack formation stage behavior. On the other hands, the stiffness decreased to approximately 8% or lower than the initial stiffness by cracking in all carbon grid-reinforced specimens, and the post-peak behavior also exhibited to be dependent on the tensile stress acting on carbon grids under the maximum load based on 80% of the tensile strength. ”
3. We appreciate the comment. The size of the carbon grid-reinforced specimens was determined in order to occur the flexural failure governed by mortar crushing considering the universal testing machine condition. The size of the steel-reinforced specimen was also determined to have the similar tensile capacity to the specimens reinforced with both the biaxial wrap knitting and cross-laminate structured carbon grids. Sentences in section 2.1 were added to enhance the description of these things as follows;
“The size of the carbon grid-reinforced specimens (Figure 2(a), 2(b) and 2(c)) was determined so that the failure mode was governed by mortar crushing considering the universal testing machine (UTM) condition. The size of the steel-reinforced specimen (Figure 2(d)) was also determined so that the tensile capacity by reinforcement was similar to the specimens reinforced with both the biaxial wrap knitting and cross-laminate structured carbon grids.”
Also, the tensile properties of steel and the compressive strength of mortar were added in Table 2 and Table 3, respectively .
4. We appreciate the comment. We added the sketch of a test set-up in Figure 5. In Figure 2, the details of carbon grid arrangement for all specimen were also provided for ease understanding of the carbon grid arrangement.
5. We appreciate the comment. We reviewed and corrected the grammatical and language errors.
6. We appreciate the comment. We extensively revised and added the conclusion section as below considering what you recommended.
“ 4. Conclusions
In this study, the flexural behavior of TRC one-way slabs was evaluated by performing a three-point bending test on nine carbon grid-reinforced specimens and a steel-reinforced specimen. Carbon grids manufactured by two different methods with varying cross-sectional areas, spacings, tensile strengths, and elastic moduli of longitudinal strands were incorporated into the specimens. The study findings can be summarized as follows.
- The specimens reinforced with the biaxial warp knitting structured carbon grids exhibited smaller crack spacings and more cracks than the specimens reinforced with the cross-laminate structured carbon grids. This was considered to be an effect of smaller strand spacing of the biaxial warp knitting structured carbon grid than the cross-laminated structured carbon grid.
- The biaxial warp knitting structured carbon grid had smaller cross-sectional areas and spacings of strands but higher tensile strength than the cross-laminate structured carbon grid. Consequently, the specimens reinforced with the biaxial warp knitting structured carbon grids had the lower tensile capacity than the specimens reinforced with the cross-laminated carbon girds. However, the maximum load had a tendency to be higher in the specimens reinforced with the biaxial warp knitting structured carbon grids than the specimens reinforced with the cross-laminate structured carbon grids even though the tensile capacity was similar or low. Specimen G2 which had the highest tensile capacity among the specimens reinforced with the biaxial warp knitting structured carbon grids also exhibited a maximum load of 1.10 times higher than specimen J5 which had the highest tensile capacity among all carbon grid-reinforced specimens.
- All specimens reached the maximum load simultaneously with mortar crushing, and little damage occurred to the cross-laminated structured carbon grids until the end of the experiment. In particular, the specimens reinforced with the cross-laminated structured carbon grids seems to have used excessive carbon grids compared to the specimen conditions such as the effective depth and compressive strength of mortar.
- When the maximum load was reached, strains corresponding to 38–82% of the tensile strain were observed on the longitudinal strands of the carbon grids. Then, the load decreased as the load resisted by the mortar was transferred to the carbon girds right after the peak. The flexural strengths calculated based on the force equilibrium and strain compatibility equations was also found to conservatively evaluate the maximum loads by the experiment.
- The load decreased right after the maximum load, but the load reduction was stopped or the load slightly increased with the increasing mid-span deflection in the case of the specimens in which a stress of less than 80% of the tensile strength was applied to the longitudinal strands under the maximum load. The experimental results validate that a design for generating tensile stress of a carbon gird that corresponds to less than 80% of the tensile strength at the ultimate state and securing bond between a carbon grid and mortar is required to prevent brittle failures right after reaching the maximum load caused by the carbon grid fracture.
- Among the specimens reinforced with the cross-laminate structured carbon grids, the specimens reinforced with the stiffer carbon grids because of the largest cross-sectional areas or small spacing of strands immediately exhibited the behavior of the crack stabilization stage without the crack formation.
- Except the stiffer carbon grid-reinforced specimens, the stiffness of the carbon grid-reinforced specimens after cracking decreased to approximately 8% or lower than those before cracking. Moreover, the stiffness did not significantly increase in the crack stabilization stage compared to that observed in the crack formation stage, thereby the load and mid-span deflection relationship might be modeled as a bilinear curve where the stiffness changes around the crack initiation point. Meanwhile, the calculated stiffness of the carbon grid-reinforced specimens after cracking based on the existing equation of the effective moment of inertia for TRC members revealed to be 1.5 times higher than the experimental results because the equation was proposed based on TRC members reinforced with GFRP bars which had lower tensile modulus and strength than the carbon grids in this paper.
- Although the carbon grids with higher tensile force than steel were arranged in the specimens, their maximum load ranged from 60~110% of the maximum load of the steel-reinforced specimen. Additionally, the stiffness of the carbon grid-reinforced specimens after cracking was significantly lower than that in the steel-reinforced specimen. Although the steel-reinforced specimen exhibited ductile behavior, the carbon grid-reinforced specimens exhibited kind of brittle behavior, wherein the load sharply decreased in stages as the load resisted by the mortar was transferred to the carbon grids.
- The test results were drawn by nine one-way slab specimens with two different types of carbon grids and mortar with one level of compressive strength. Consequently, further researches considering various levels of compressive strength, the effect of transverses strands, the prediction of cracked section stiffness and so on are considered to be required.”

Round 2
Reviewer 2 Report (New Reviewer)
The current version can be accepted.
Reviewer 3 Report (New Reviewer)
comments are done
This manuscript is a resubmission of an earlier submission. The following is a list of the peer review reports and author responses from that submission.
Round 1
Reviewer 1 Report
This paper deals with the flexural behavior of TRC one-way slabs with different types and amounts of grid-type CFRP reinforcement impregnated in resin. An experimental campaign has been performed and the results have been compared with a reference specimen of RC. The conclusions have low reference value and there are some flaws, described in the lines below. This lead to the recommendation of the present reviewer to reject the manuscript:
1.-It is not clear which are the main contributions of the paper to the scientific literature in this field.
2.-The discussion of the results is simple, no analytical study has been provided and neither a comparisons with other results in the literature. This leads to not surprising conclusions.
3.-In the literature there are some valuable references that analyze the bending behavior of TRC, which are not mentioned or used as reference in study presented in the manuscript, for instance:
Williams Portal, N., Nyholm Thrane, L. & Lundgren, K. Flexural behaviour of textile reinforced concrete composites: experimental and numerical evaluation. Mater Struct 50, 4 (2017). https://doi.org/10.1617/s11527-016-0882-9
4.-No references are provided that study and validate the use of strain gauges in CFRP grids impregnated in resin.
5.-The manuscript there is no discussion of the absence of plastic redistribution in CFRP, which allows in conventional RC to reach the full capacity of rebars.
6.-It would have been helpful to include specimens only reinforced longitudinally to compare their behavior to the specimens reinforced with a grid.
7.-Names of specimens vary along the paper, in some Figures they are referred by the carbon grid denomination (for instance Figures 9, 10 and 18), instead of the specimen name. This is not helpful for the reader.
8.-Test configuration and data are not fully described. As an example, concrete mortar properties, test control procedure (force/displacement) or the model of LVDTs employed are not defined.
9.- A brief description of carbon grids biaxial warp knitting and cross-laminate structures would have been helpful for the readers.
10.-The authors must improve the writing and the use of English. In the following lines, there are some examples to illustrate this statement:
· Line 10: “TRC”, avoid using acronyms, not mentioned before, in the abstract.
· Line 42: might be improved by adding “polymers (FRPs) with strong resistance to corrosion when used as the reinforcement for concrete structures”.
· Line 187: “crack spacing” should have been employed instead of “crack spaces”
· Line 199: might be improved by changing “occurred” with “developed”.
· Line 200: might be improved by changing “under the transformation” with “due to transfer”.
· Line 283: Please it rephrase “stage-wise load reduction tendency”, residual load capacity seems to be related to the deformation imposed on the specimen.
Reviewer 2 Report
Flexural behavior of one-way slab reinforced with grid-typeCFRPs” The article is interesting. A few observations are given below,
1) An abstract is a short summary of your completed research. It is intended to describe your work without going into great detail. Abstracts should be self-contained and concise, explaining your work as briefly and clearly as possible.Objectives and aims are missing in the abstract. Further, theabstract should provide an overview of proposed methods/methodology, materials with obtained results in the form of quantitative values.2) Some more latest studies are required in the introduction section to further highlight the importance of this study.Following article is include for the reference of the authors to improve the introduction section.
• Saleem, M. U., Qureshi, H. J., Amin, M. N., Khan, K., & Khurshid, H. (2019). Cracking behavior of RC beams strengthened with different amounts and layouts of CFRP. Applied Sciences, 9(5), 1017.
3) For readers to quickly catch the contribution in this work, it would be better to highlight major difficulties and challenges, and authors' original achievements to overcome them, in a clearer way in Introduction section.
4) please provide the legends in each graph. (e.g., figure 13 legends are missing)
5) It is suggested to highlight the limitations of this study, suggested improvements of this work and future directions in the conclusion section. Also, the conclusion can be presented better than the present form with more findings.